# Development of Systemic Interventions to Decrease Breast Cancer Risk: A Group Concept Mapping Study

**DOI:** 10.3390/ijerph21030318

**Published:** 2024-03-09

**Authors:** Janet Gray, Carrie Petrucci, Connie Engel, Nyisha Green-Washington, Nancy Buermeyer

**Affiliations:** 1Breast Cancer Prevention Partners, San Francisco, CA 94109, USAnyisha@bcpp.org (N.G.-W.); 2Department of Psychological Sciences, Program in Science, Technology, and Society (STS), Vassar College, Poughkeepsie, NY 12604, USA; 3Concept Systems, Inc., Ithaca, NY 14850, USA; carrie.petrucci@gmail.com

**Keywords:** breast cancer, primary prevention, community-partnered participatory research, group concept mapping, community public health

## Abstract

As breast cancer continues to take a devasting public health toll, most primary prevention approaches are targeted at individual actions. We have proposed, instead, developing systemic, population approaches to preventing the disease. We used a combined qualitative–quantitative methodology, group concept mapping (GCM), to identify Importance and Feasibility ratings of systemic interventions across a wide spectrum of approaches and stakeholders. Participants (*n* = 351) from across the state of California sorted 84 potential interventions into topical piles, and then rated each intervention on perceived Importance and Feasibility. Multidimensional scaling and a cluster analysis identified eleven clusters or themes of interventions. Participants rated interventions on Importance and Feasibility differently depending on the region of the state in which they lived. The results of this study underscore the importance of sharing health information with and seeking public health solutions from community partners in general and from beyond the urban areas usually studied.

## 1. Introduction

### 1.1. Background

In the United States, cancer of the breast is the leading form of female cancer. The American Cancer Society estimates that 297,790 women (and 2800 men) will have been diagnosed with the disease in 2023 and 43,170 women (and 530 men) will have died from it [1]. In California, more women are diagnosed and die from breast cancer than from any other state [1].

Breast cancer is a complex set of disorders with several different subtypes, affecting diverse groups of people at different rates and in different ways. For example, although the incidence rate for Black women in the US is just 1% lower than that of white women, Black women currently die from breast cancer at a 40% higher rate than their white counterparts [1]. Young Black women are diagnosed significantly more often than young white women with triple-negative breast cancer, a particularly aggressive form of the disease. This difference is increased significantly for young Black women from lower-resourced and socio-economically challenged neighborhoods [2]. Recent data indicate that Black women with a diagnosis of breast cancer before the age of 40 had a significantly higher rate of recurrence than did white women of the same age. Older (>55) Black women diagnosed with the disease had the worst overall survival rates compared with younger and white women [3].

The complexity of the disease is also reflected in the strong evidence that there are a range of risk factors for breast cancer, many of which interact with one another in affecting the risk of developing the disease for individual women. These risk factors are diverse, including factors associated with the social and built environment; exposures to environmental toxicants; diet, nutrition, and physical activity; tobacco and alcohol use; age and reproductive history; genetics and family history; and several other factors [4,5,6,7,8].

Scientists estimate that more than 50 percent of all breast cancer is preventable at a population level [9]. Yet most intervention strategies are focused on individualized practices such as increased exercise, better diet, and decreased exposures to environmental toxicants, regardless of whether these approaches are feasible for the person. Instead, we have proposed that interventions that address the *systemic root* of the issues underlying risk factors hold the greatest potential for dramatically decreasing risk for developing breast cancer across the population [4].

### 1.2. Paths to Prevention: The California Breast Cancer Primary Prevention Plan

The current project is a follow-up to a major initiative funded by the California Breast Cancer Research Program (CBCRP), a state-run program that funds research and policy work aimed at treating and reducing breast cancer burden. Its particular focus is on the role of environment and lifestyle, while also addressing disparities, inequalities, and underserved populations in California. The earlier project was undertaken by Breast Cancer Prevention Partners (BCPP), a national organization whose mission is to work toward eliminating environmental exposures linked to breast cancer risk through educational, market-based, and legislative/regulatory initiatives. The directive of the first initiative, completed in 2020 and titled *Paths to Prevention: The California Breast Cancer Primary Prevention Plan*, was to develop a statewide action plan that used the best science and community wisdom to address policy issues concerning possible systemic changes to prevent breast cancer [4]. Importantly, this plan was different from most major state cancer plans as it focused on primary prevention rather than disease detection or secondary prevention. It identified systemic, societal-level interventions, rather than individual behaviors, as strategies to reduce population-level risk for developing breast cancer. The full project was considered through a social justice lens, adhering to the goals and principles of a community-partnered participatory approach [10,11].

The process of *Paths to Prevention* involved a full scoping review of the scientific literature relevant to 23 categories of risk factors. This scientific review process was conducted in parallel and in conversation with members of an advisory group and topical virtual study groups and in listening sessions in communities across the state of California. Of particular importance was the engagement of voices of people who are socially and economically marginalized, and often under-represented and under-valued in science and public policy discourse [11,12,13].

The current follow-up project, also funded by CBCRP and led by BCPP, examines numerous possible systemic interventions for the 15 factors identified in the original prevention plan as being associated with the most substantive scientific support for links with the disease: race, power, and inequalities; social and built environments; alcohol; breast feeding; chemicals in consumer products; diet and nutrition; ionizing radiation; light at night; non-ionizing radiation; occupation; pharmaceutical hormones; physical activity; place-based chemicals; pregnancy-related factors; and tobacco. ‘Race, power, and inequities’ and ‘social and built environments’ are foundational factors and exert effects on all others [4].

### 1.3. Aims and Objectives of the Current Project

Group concept mapping (GCM) is a mixed methods participatory research approach that prioritizes building knowledge from a diverse group of stakeholders, and synthesizing what is learned into a set of common themes or activities that can then be put into action [14,15]. We used GCM to (a) identify approaches that groups of stakeholders might use to form the basis of systemic intervention initiatives to be funded in a subsequent round of grant proposals to CBCRP, and (b) to engage multiple stakeholders across the state about their priorities, concerns, and capabilities to engage further in this preventive work.

## 2. Materials and Methods

### 2.1. Advisory Group

Before beginning the project, a 19-member advisory group (AG) was formed. Members were drawn from self-identified stakeholder groups including academics/researchers, healthcare practitioners, policy experts/advocates, and community members from across many regions of the state of California. AG member discussions provided early information on priorities, resources, and means of reaching out to broader and diverse communities across the state as the participant recruitment stage began.

### 2.2. Development of the Target Interventions for the GCM Exercise

We began with almost 400 potential intervention topics that emerged from the initial *Paths to Prevention* project that reviewed the science and proposed possible interventions (see Figure 1). This list was culled based on duplication and lack of appropriateness, e.g., emphasis on personal rather than systemic interventions; unrealistic goals given the limitations of the resources available to participants; etc. [15]. Remaining suggested interventions were then subjected to review of existing evidence supporting the likelihood that an intervention would be effective in decreasing exposure to a particular risk or set of risks. This was accomplished through searching public-health-based databases (e.g., Healthy People 2020), systematic reviews (e.g., Cochrane Reports), peer-reviewed literature gleaned from academic databases (e.g., SCOPUS, PubMed, PsycINFO), and descriptions of relevant existing programs that included assessment.

This process decreased the number of potential intervention possibilities to 141. Three members of the BCPP research staff, in collaboration with the GCM statistical consultant and in conversation with the AG, worked to reach consensus on the final 84 interventions. This process involved dropping interventions without adequate evidence to support their efficacy (3 interventions), eliminating interventions that reflected existing laws (3 interventions), and cutting those that were redundant or lacked clarity (51 interventions) (Figure 1). The final 84 interventions were spread across the 15 risk factors of interest as well as perceived feasibility of engagement in some version of the intervention.

### 2.3. Participants

Participants were members of four self-identified stakeholder groups (academics/researchers, healthcare practitioners, policy experts/advocates, and community members) from across the 8 main regions of the state of California (see Table 1). They were recruited through several mechanisms, including (but not exclusively) (1) members of the project advisory group, (2) members of allied organizations (environmental health and environmental justice organizations; groups promoting breastfeeding, food justice, unions, reproductive justice; groups especially concerned with health and environment in communities of color and other marginalized groups, tribal organizations, etc.), (3) academic researchers on the BCPP Science Advisory Panel or with whom BCPP members had collaborated on other research projects, as well as the CBCRP researcher list, and (4) public health school postings. Interested parties were directed to an online description of the project and the tasks to be performed (see below). If interested, potential participants completed a Google contact sheet. They were also requested to ask members of their respective networks and communities to consider participating.

There were three parts to the online process (see below). A total of 351 participants completed at least one of the GCM tasks, with 340 completing the sorting process, and 337 and 334 participants, respectively, completing the Importance and Feasibility ratings.

Participants were offered a USD 50 gift card for completing the GCM tasks. Members of the AG were offered USD 500 honoraria, in recognition of their ongoing engagement and support throughout the project’s timeline.

All participants completed an online consent form. After GCM tasks were completed, all data were de-identified and analyses were completed anonymously.

### 2.4. GCM Data Collection and Analysis

To carry out the online data collection and analysis, we used a group concept mapping (GCM) approach in tandem with the Concept System® groupwisdom^TM^ software (Build 2022.24.01 [web-based platform], Concept Systems, Inc.; Ithaca, NY, USA). GCM is a mixed methods framework that first gathers qualitative data in the form of a large number of responses or statements to an open-ended question [14]. Multivariate analyses, consisting of multidimensional scaling [16] and a hierarchical cluster analysis [14,16,17], are then performed within the *groupwisdom*^TM^ software, which produces visuals including cluster maps, pattern matches (ladder diagrams), and go-zones (see below) that aid in the analysis, interpretation, and utilization [14,15].

#### 2.4.1. Online Data Collection

The online data collection process was open for almost four months. After providing basic demographic information (stakeholder group, age, racial/ethnic identity, gender identity, residential region within CA, and whether they were living with a diagnosis of breast cancer), participants were asked to sort electronically each of the 84 interventions into categories that made sense to that individual. These “piles” were then named by the participant with common themes for the interventions [18]. This sorting task was used in the initial multidimensional scaling and cluster analysis to identify themes or ways to reduce breast cancer, as identified across the participants who completed the sorting task, and which met the criteria of a quality assurance assessment process [15].

The second part of the online project involved responding with ratings for each of the 84 interventions to the prompt “one way to reduce breast cancer risk, particularly in disproportionately affected communities, is….” The responses involved coding on a scale of 1 to 6 (6 representing the highest level), first for Importance and then for Feasibility.

#### 2.4.2. Multidimensional Scaling and Hierarchical Cluster Analysis

Sorted data were used to create an 84 × 84 matrix of similarities, with each cell containing the frequency by which two intervention statements had been sorted into the same pile. A ‘point map’ was generated using non-metric multidimensional scaling (MDS), with each point on the 2-dimensional map representing the likelihood of statements having been sorted into the same theme. Statements that were more commonly sorted together appeared closer on the map [15]. The MDS analysis of the similarity matrix converged after 10 iterations.

Validity and reliability of the sort and rating data were assessed in multiple ways, including strategies developed specifically for concept mapping by Trochim [19] and refined by Rosas and Kane [20].

The X–Y coordinates produced from the point map were then subjected to a hierarchical cluster analysis using Ward’s method [17,19]. Each statement began in its own cluster (an agglomerative approach) and was then combined with statements with which it was closest on the map. This produced a number of non-overlapping cluster solutions of potential interventions. After iterative running of the cluster analyses, the most parsimonious solution included 11 clusters. Finally, four higher-level themes were identified by the research team by evaluating commonalities across clusters with close proximity on the final map.

#### 2.4.3. Pattern Match/Ladder Analyses

Qualitative pattern match or ladder analyses presented visual comparisons of mean cluster ratings between main rating variables (Importance and Feasibility), and across these variables as broken down by stakeholder grouping and state region [15].

#### 2.4.4. Go-Zone Maps

Two-dimensional go-zone maps provided visual profiles of where each potential intervention fell with respect to mean values for Importance and for Feasibility. Four areas of the map indicated those interventions that were rated above average (mean) for both Importance and Feasibility, those that were above average for Importance but below average for Feasibility, those that were below average for Importance but above average for Feasibility, and those that were below average for both ratings [15].

Go-zone quadrants were examined for both the entire dataset as well as for individual clusters. For regional analyses, each intervention was scored for the number of regions that rated it in each of the quadrants, and averages across the interventions for a given region were calculated.

## 3. Results

### 3.1. Demographics of Study Participants

As shown in Table 1, over half the participants (55%) self-identified as community members, while fewer self-identified as healthcare practitioners (24.2%), academic/research professionals (10.5%), or policy experts/advocates (9.7%). Although there were representatives from each of the eight regions of California, the highest numbers came from the San Francisco Bay (21.4%) and northern California (19.7%) regions and the fewest from the Central Valley (6.8%), Inland Valley (7.1%), Central Coast (8.0%), and San Diego/Orange County (8.6%) regions.

Over half (55%) of the participants identified as white, with 18.8% identifying as Black/African American and 13.7% as Latina/Hispanic/Latinx. Less than 4% of the total participant pool self-identified for each of the other categories. The majority (63% of the participants) were between the ages of 26 and 40, with no one younger than 26 years and about a third being older than 40 years.

In addition to the 78.4% of participants who identified as a female/woman, 18.8% identified as a male/man and 0.6% as gender non-conforming. Just over a quarter of respondents (26.8%) were living with a diagnosis of breast cancer, while 72.9% were not.

### 3.2. Cluster Analysis

The hierarchical cluster analysis resulted in the identification of 11 clusters of potential interventions and four themes exploring means by which interventions could be implemented (Figure 2).

Appendix A includes the full set of individual interventions, organized by cluster.

Higher-order thematic quadrants responsive to the focus prompt were designated as Q1: through legal action at the local or state level, Q2: through access to healthy food and exercise, Q3: through education and advocacy in the community, and Q4: through best practices and regulations in the workplace and in communities (Figure 2).

### 3.3. Validity and Reliability of Cluster Analysis Results

Strong validity was found for the sort data from the multidimensional scaling based on a stress value (0.24) that assesses model fit [19,20] and by a measure of configural similarity (0.524) [16,20]. Strong reliability of the individual and aggregated similarity and distances’ matrices was found based on five reliability measures. These included running correlations between the two aggregated similarity matrices for split halves of the data (0.977), the two aggregated distance matrices for split halves of the data (0.970), the averaged individual sort matrices and the total aggregated similarity matrix (0.954), the averaged individual sort matrices and the total aggregated distance matrix (0.921), and the averaged correlations between all of the individual sort matrices (0.840). The Spearman–Brown correction was applied to these five reliability indices to adjust for only a partial sample being utilized in the correlations [16,20]. For the last three reliabilities based on individual sort data, due to the high number of total sorts (340), a random sample of 40 sorts (5 sorts from each of the eight regions included in the study) was selected to avoid inflation of reliability estimates.

Internal consistency for both Importance and Feasibility ratings was strong based on Cronbach’s alpha of 0.98 for each (*n* = 217 for Importance and *n* = 204 for Feasibility). To measure inter-rater reliability, intraclass correlations were used for both ratings, and were also within acceptable ranges [20] at 0.975 for Importance ratings and 0.978 for Feasibility ratings. In summary, strong validity and reliability of the similarity data, the distance data, and the ratings were found, with results falling within established cut-offs [16,20].

### 3.4. Relative Ratings of Mean Values for Importance and Feasibility across Clusters

Figure 3 shows the relative ratings of mean values for Importance (Figure 3a) and Feasibility (Figure 3b) for each cluster. More layers indicate higher quintiles of the respective rating scores. Those clusters with the highest mean ratings for Importance were found in the themes that entailed regulatory or legal action. These included ‘regulate hazardous products’ (Cluster #11), and ‘promote policies to reduce alcohol and tobacco use’ (#10), followed by ‘require transparency and best practices to reduce harmful exposures’ (#9), ‘reduce hazardous workplace exposures’ (#8), and ‘implement preventive health measures’ (#7). Rated lower, but still relatively highly on the scale of 1–6, were ‘promote community self-determination’ (#1) and ‘create interventions related to reproductive and women’s health’ (#6), followed by ‘provide preventive education and resources’ (#5), ‘incorporate healthy lifestyle activities in schools and communities’ (#4), ‘increase access to healthy food and exercise’ (#3), and ‘reduce air, water, and light pollution’ (#2).

Feasibility ratings were highest in the two quadrants that engaged community action: ‘through education and advocacy in the community’ and ‘through best practices and regulations in the workplace and in communities.’ Highest mean cluster ratings for Feasibility were found for ‘implement preventive health measures’ (#7) and ‘create interventions related to reproductive and women’s health’ (#6), followed by ‘incorporate healthy lifestyle activities in schools and communities’ (#4) and ‘promote policies to reduce alcohol and tobacco use’ (#10). Slightly lower average feasibility ratings were found for ‘provide preventive education and resources’ (#5), ‘require transparency and best practices to reduce harmful exposures’ (#9), and ‘reduce hazardous workplace exposures’ (#8), followed by ‘regulate hazardous products’ (#11), ‘increase access to healthy food and exercise’ (#3), and ‘promote community self-determination’ (#1). The cluster that was perceived as being least Feasible, on average, was ‘reduce air, water, and light pollution’ (#2) (Figure 3b).

### 3.5. Pattern Match/Ladder Analyses

Using pattern match (ladder analysis) comparisons, mean ratings for each of the 11 clusters were visualized for Importance and Feasibility for the full universe of participants and also broken down by stakeholder group and region of the state. These pattern match figures are purely visual and are not subject to an inferential statistical analysis. Yet, they clearly demonstrate the relative Importance or Feasibility of different intervention clusters as articulated by participants with different demographic characteristics.

Figure 4 visually presents the differences in absolute Importance and Feasibility mean ratings for each cluster using the data provided by all participants statewide (334 Importance ratings and 327 Feasibility ratings). All mean ratings were quite high, ranging between 4.37 and 4.99 on a scale from 1–6, with 6 being the highest. The ordinal difference in ratings of the clusters for the two rating scales is indicated by the crossing of lines between the two y-axes, with a correlation (Pearson r) of 0.39 indicating low association between the overall Importance and Feasibility ratings of interventions across the clusters. For example, while ‘regulate hazardous products’ was rated most highly on Importance across all participants, it fell into the bottom half of Feasibility ratings. And while ‘promote community self-determination’ was rated third highest for Importance, it was rated third lowest for Feasibility (Figure 4).

Breaking down the full dataset by stakeholders yields a more complicated picture (Figure 5). While members of all stakeholder groups rated the ‘regulate hazardous products’ cluster with their highest scores for Importance, all groups except healthcare practitioners rated the Feasibility of implementing this cluster of interventions in the lower three of the cluster categories. The ‘promote community self-determination’ cluster ratings for Importance were relatively high within the ranges of scores from all stakeholders, especially for academic/research professionals. Ratings for Feasibility of this cluster of interventions, however, were much lower across the four categories, but especially for community members and healthcare practitioners. Correlations for ratings of Importance across the 11 intervention clusters ranged from 0.92 for advocates and academics to 0.66 for advocates and healthcare practitioners. There was a greater spread in ratings for Feasibility of interventions, with correlations ranging from 0.93 for advocates and academics to 0.26 for academics and healthcare practitioners (see Figure 5a,b).

Pattern match diagrams comparing regional average ratings for both Importance and Feasibility show even more complex patterns (see Figure 6a,b). Correlations for intervention cluster ratings for Importance run from 0.98 for San Diego/Orange counties and the Los Angeles region to 0.15 for the Central Coast and Central Valley regions. Participants from all regions rated the ‘regulate hazardous products’ cluster relatively high in Importance and the ‘reduce water, air, and light pollution’ relatively low in these ratings. The remainder of the ladder has many crossing rungs, indicating differences in priorities across regions (Figure 6a).

Correlations for Feasibility ratings for intervention clusters across the eight regions were also quite dissimilar, with the urban regions of San Francisco Bay and Los Angeles having a high (0.96) Pearson r value while the Central Coast and northern California regions had a correlation of only 0.08 in the respective participant ratings of Feasibility of different intervention clusters. Although participants from all regions of the state rated ‘reduce water, air, and light pollution’ as the lowest cluster in Feasibility, there was considerable variability in the ordinal positioning of all other clusters across the regions of the state (Figure 6b).

### 3.6. Go-Zone Maps

Go-zone maps present comparisons for individual points of the relative average Importance vs. Feasibility ratings for individual interventions or clusters of interventions [15]. Figure 7 presents the go-zone map for all potential interventions as rated by all participants. The upper right, green quadrant holds those interventions that were deemed above average (mean) for both Importance and Feasibility, while the lower left, grey quadrant contains those interventions that were rated as below average for both Importance and Feasibility. The other two quadrants have points representing interventions with mixed high/low ratings, although always within the context of all ratings being high overall, given the rating scale of 1–6. Correlation values indicate the relative relationship between Importance and Feasibility ratings across the interventions of the particular go-zone map (Figure 7).

Individual interventions from each of the 11 clusters were positioned in both the green (high Importance and high Feasibility) and grey (low Importance and low Feasibility) quadrants, although not all clusters were represented in the mixed rating quadrants. See Appendix A for lists of all interventions, organized by cluster, and with go-zone placement information for each intervention.

As an example, Figure 8 offers a go-zone map for one of the smaller clusters (Cluster 4), ‘incorporate healthy lifestyle activities in schools and communities’. Two proposed interventions were rated high on both Importance and Feasibility: ‘provide free and healthy breakfast and lunch programs without income requirements to K–12 students when school is in session and during summer breaks’ and ‘work with school districts to ensure K–12 schools have the resources they need to offer niversal physical education classes’. Three intervention proposals were rated as low (below mean) in both Importance and Feasibility: ‘incorporate student input to provide healthy school lunch programs that students are likely to eat’, ‘collaborate with and fund local organizations to provide physical activity opportunities in after school, camp, or recreation programs’, and ‘increase free physical activity options in underserved communities by creating culturally tailored programs across the lifespan (for example, walking programs and Zumba in the park)’. The other two interventions had mixed (high/low) ratings (Figure 8).

Looking at go-zone maps across the 11 clusters supports the findings of the initial cluster rating analyses (individual maps not shown.) Those clusters whose interventions were found in the two high-Importance quadrants, across the range of clusters, were more likely to fall under the legal and regulatory intervention theme (quadrants 1 and 4 in Figure 2).

Ratings from go-zones for each region (individual maps not shown) of interventions in the various quadrants are presented in Appendix A, along with the average number of interventions falling in each quadrant for each cluster. For all interventions, across all quadrants, r = 0.52 for Importance vs. Feasibility ratings.

On average, interventions in the ‘regulate hazardous products’ cluster were rated as being of high Importance by 7 of the 8 regions, followed by ‘implement preventative health measures’ (6 of 8), ‘promote policies to decrease tobacco and alcohol use’ (5.5 of 8), ‘promote community self-determination’ (5 of 8), and ‘decrease hazardous workplace exposures’ (4 of 8). Interventions for all other clusters were rated of high Importance, on average, by respondents from less than half of the regions (Appendix A).

Separating the high Importance go-zone quadrants into high Feasibility (upper right quadrant) vs. low Feasibility (lower right quadrant) demonstrated that although respondents across many regions rated ‘regulate hazardous products’ and ‘promote community self-determination’ relatively highly on Importance, combined high Importance and high Feasibility ratings were relatively low (3.7 and 1.8, respectively) for the interventions in the two clusters (Appendix A).

For 9 of the 11 clusters, participants in fewer than half the regions rated the average responses to particular interventions as being both high in Importance and high in Feasibility (see Appendix A). There were two clusters for which there were average ratings across the interventions as high in both Importance and Feasibility in more than four regions, but none reached the level of high–high ratings in five or more regions (Appendix A).

## 4. Discussion

GCM is a mixed qualitative–quantitative methodology that elicits responses from many broadly recruited individuals, produces a group-constructed framework of related concepts, and allows for comparisons of priorities among groups of participants [15]. The results can provide the basis for establishing parameters for future research and active collaborative engagement on themes that emerge through the GCM process.

This project used GCM to identify 11 clusters of potential systemic interventions, all conceived in response to the prompt “one way to reduce breast cancer risk, particularly in disproportionately affected communities, is….” Clusters were further identified as addressing one of four major means by which potential interventions might be implemented. The diversity of the 84 proposed interventions, and the means for realizing them, intentionally allows different approaches for the future multidisciplinary implementation of efforts to address systemically each of the categories of risk factors identified previously in the *Paths to Prevention* report [4].

For most clusters, average Importance ratings were higher than those for Feasibility, although the trend was reversed for interventions associated with the theme of providing education and advocacy in the community (‘create interventions related to reproductive and women’s health’, ‘provide preventive education and resources’, and ‘incorporate healthy lifestyle activities in schools and communities’).

Strikingly, in the combined (all participants) absolute pattern match for Importance and Feasibility (Figure 4), ratings for Importance were highest for the ‘regulate hazardous products’ cluster, but lowest for ‘reduce air, water, and light pollution’. This suggests a dissociation between the need to regulate the final hazardous materials in products we use in our daily life and the larger environmental spheres polluted by the life-cycles of those products. One possible explanation for this apparent dissociation is that many people believe that the government, especially state and federal Environmental Protection Agencies, is already doing all that is possible to address these issues. This attitude has been expressed in both listening sessions in the original *Paths to Prevention* project and in the early regional meetings associated with the current project, even though in practice, many gaps remain in the regulation of toxic chemicals [21].

Responses on the pattern match ladders for ratings of Importance were relatively highly correlated (r = 0.66–0.92) across each pairing within the four stakeholder groups. For Feasibility, however, the healthcare practitioners’ responses stood out as being less well correlated (r = 0.26–0.48) with responses from other stakeholders. The voices of healthcare practitioners add an important dimension to the discussion, especially since they represent the source of much of the (non-internet based) information patients with breast cancer and their support networks receive related to risks for developing breast cancer and potential interventions to prevent the disease [22,23,24].

Variability in ratings of both Importance and Feasibility across regions of the state are not surprising, given the social, economic, and geographical diversity of California communities. These observations are currently being used to develop region-appropriate strategies that acknowledge cultural, racial/ethnic, economic, historic, and environmental factors that lead to differences in identity, priorities, and engagement with regulatory, community, and personal resources. We expect, in turn, that the local regional discussions that are just beginning will lead to the emergence of different priorities and proposals for moving this project forward.

### 4.1. Community-Partnered Participatory Approach

The current project was built on the principles and goals of community-partnered (-based) participatory research. Yet, there are some important distinctions that make this work unique. Most notably, the research was initiated by the staff of an environmental health/social-justice-based advocacy group, BCPP, not a group of academic researchers, as is the usual approach in community-partnered research. Much of the effort expended by more traditional researchers is in breaking down assumptions of objectivity and power dynamics between researchers and participants, with academics needing to form sustained partnerships with community members who then participate actively in all aspects of the research project [25]. In the current case, BCPP staff, who have expertise in environmental health science, policy advocacy, and community outreach, established working relationships with academic researchers and, most importantly, a broad and diverse set of stakeholders, especially from marginalized communities most affected by exposures to known risk factors for breast cancer. While issues of social dynamics and priorities still needed to be addressed as the project moved forward, these processes became essential for BCPP as it worked toward its goal of creating long-term, authentic partnerships with diverse members of disproportionately impacted and underserved communities across the state.

### 4.2. Strengths and Limitations of the Project

One strength of the current project was the intentional inclusion of participants representing different stakeholders, including a quite large and diverse set of self-identified community members. The voices of these participants added perspectives and experiences that are otherwise often missed by professionals in the field, whether academics, healthcare practitioners, or policy experts. Community members are experts in what is needed and possible, and already happening, within their own communities and can communicate best the priorities from their more local perspectives. The importance of involving community members in research and in policy development is increasingly being valued for its substantial increase in authenticity over more traditional practices [26].

There were also limitations to this study. There was no attempt to gather a random representation of stakeholders from each of the eight regions or across the state as a whole, and distribution across many of the demographics was uneven. This prevented us from a more in-depth analysis of several potential mediating effects.

And while the research team had worked diligently to make sure that various interventions were related to well-documented risk factors for breast cancer, many of these underlying factors (e.g., light at night, non-ionizing radiation, environmental exposures) were not as closely associated with breast cancer risk by some participants. This may have affected ratings of Importance and Feasibility, given that the prompt for ratings was “to reduce breast cancer risk, particularly in disproportionately affected communities”. Nevertheless, all interventions received relatively high ratings for both Importance and Feasibility on the 6-point scale, and there were reliable differences in ratings by stakeholder groups and especially by region of the state.

### 4.3. Moving Forward: Implications for Research, Policy, and Practice

As part of the original *Paths to Prevention* project, listening sessions were held across California with participants coming mainly from community groups. We are in the process of organizing and holding regional meetings associated with this current study, with conversations beginning with region-specific findings on assessed Importance and Feasibility ratings of particular interventions and intervention clusters. Especially in rural areas where outside researchers and policy makers less often travel to engage these populations, session attendees often voice gratitude for being included in the process. They are frequently ignored or passed by in efforts to scope out or plan public health interventions. Across the state, one of the responses that was commonly heard was “don’t do anything about me without me” or the historically more common “nothing about us without us” [27]. People are eager to have a voice in decisions that would impact their health and environment, and to learn more about how the two are connected. Involving community members and other stakeholders in ongoing public health research and policy development is critical, as demonstrated by the engagement of participants in the current project. Other recent papers support the notion that involvement of multiple stakeholders, especially community members, is critical to improving public health strategies and policy initiatives, while also reducing community inequalities [27,28,29].

## 5. Conclusions

This group concept mapping exercise demonstrated that participants, including community members, care about the impact of various breast cancer risk factors and that they are able to articulate and prioritize potential systemic interventions to decrease risk moving forward. Differences in rating Importance and Feasibility of various interventions across the state of California indicated that this understanding is shaped by regional factors and underscore the importance of sharing health information with and seeking public health solutions from community partners from beyond the urban areas usually studied.

In taking this community-partnered approach to identifying and implementing interventions that address breast cancer risk at a regional and systemic level, BCPP hopes to continue to engage multiple stakeholders in working toward its ultimate mission: decreasing exposures to chemicals and other factors that increase risk for developing breast cancer, as well as numerous other diseases impacted by these same factors.

## Figures and Tables

**Figure 1 ijerph-21-00318-f001:**
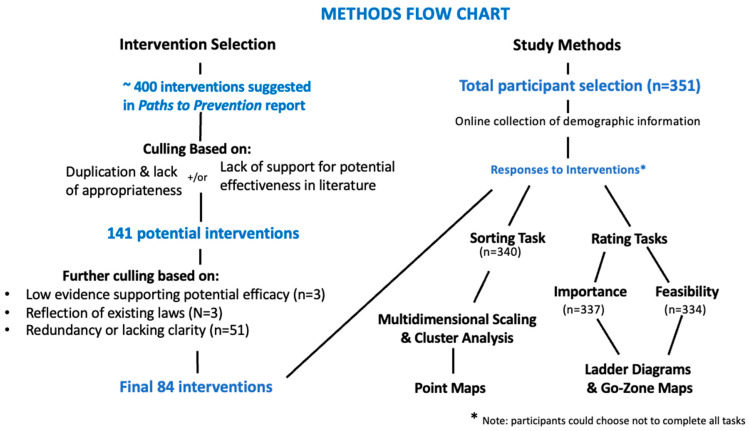
Flow chart of study intervention selection process (left side) and overall methods (right side).

**Figure 2 ijerph-21-00318-f002:**
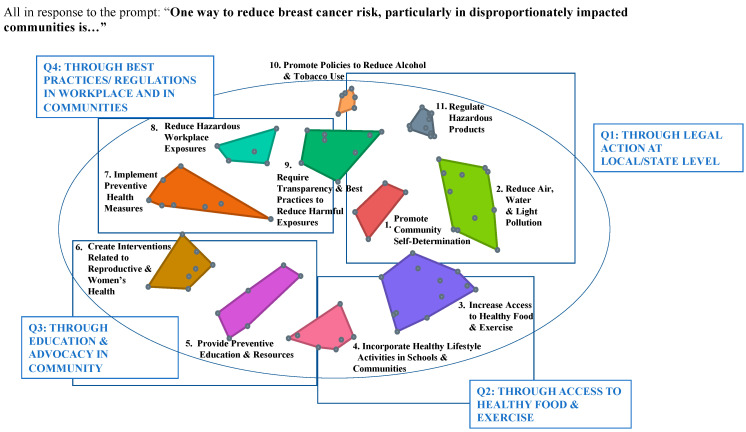
Eleven clusters resulting from hierarchical cluster analysis, set within the 4 higher-order themes, all in response to the prompt “one way to reduce breast cancer, especially in disproportionately impacted communities, is …”. Points at the vertices or within geometric shapes represent placement of individual statements on a two-dimensional map. Points that are closer to one another represent intervention statements that were more likely to have been sorted together. Colors for particular clusters are used in Figure 3, Figure 4, Figure 5 and Figure 6 also.

**Figure 3 ijerph-21-00318-f003:**
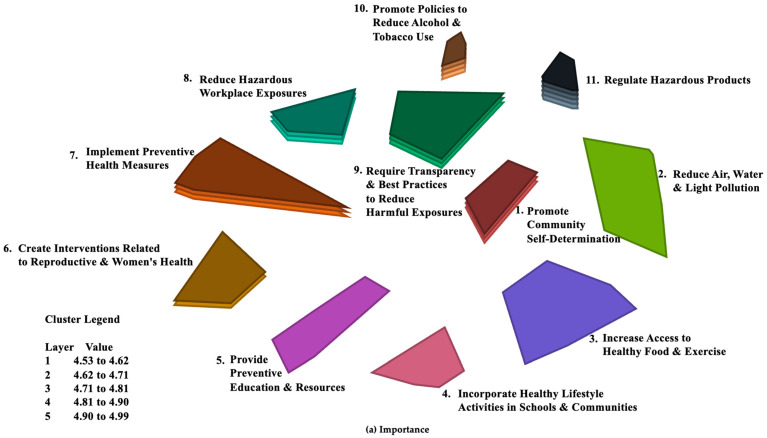
Final 11 clusters with layers denoting relative strengths of each cluster for (**a**) Importance and (**b**) Feasibility.

**Figure 4 ijerph-21-00318-f004:**
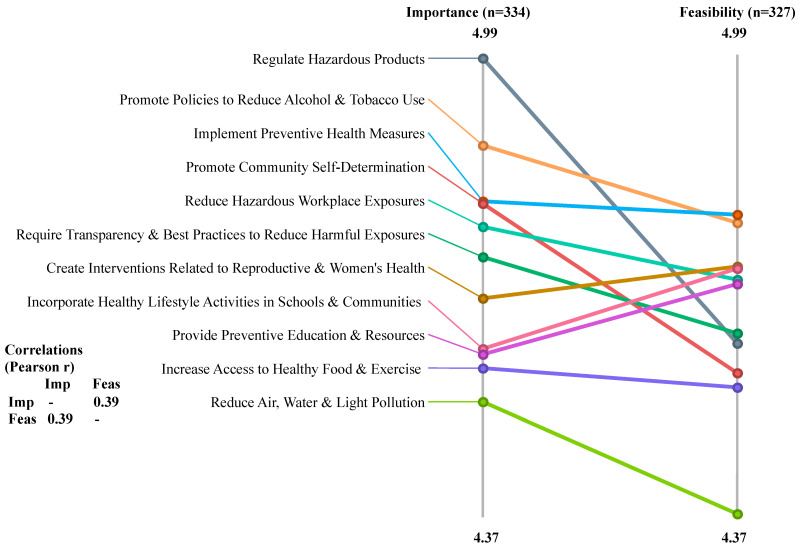
Ladder diagram of Importance vs. Feasibility for all clusters and all participants.

**Figure 5 ijerph-21-00318-f005:**
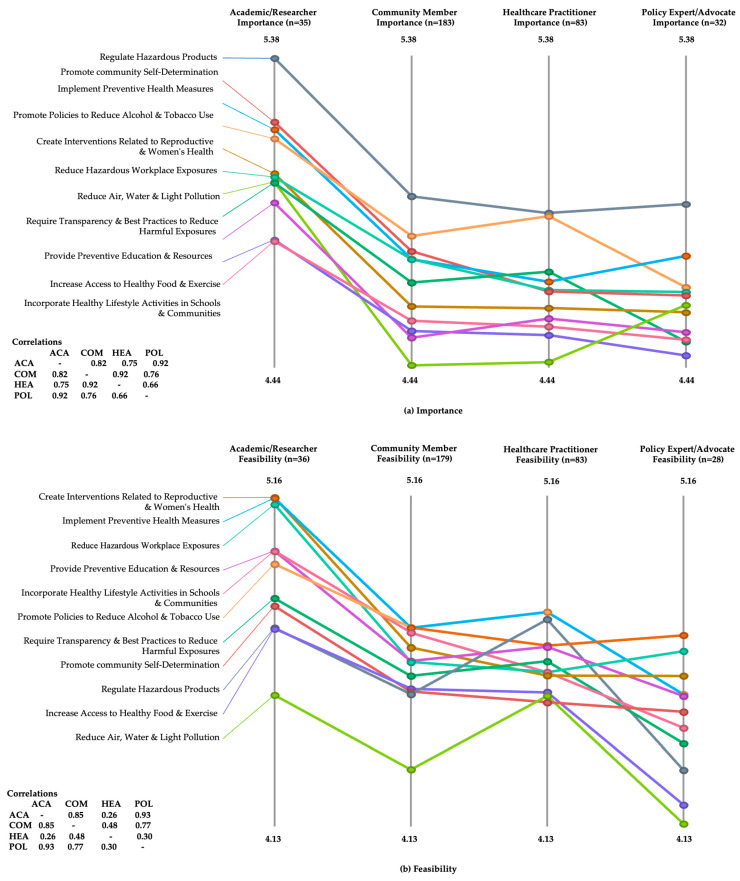
Ladder diagram by stakeholder groups for (**a**) Importance and (**b**) Feasibility.

**Figure 6 ijerph-21-00318-f006:**
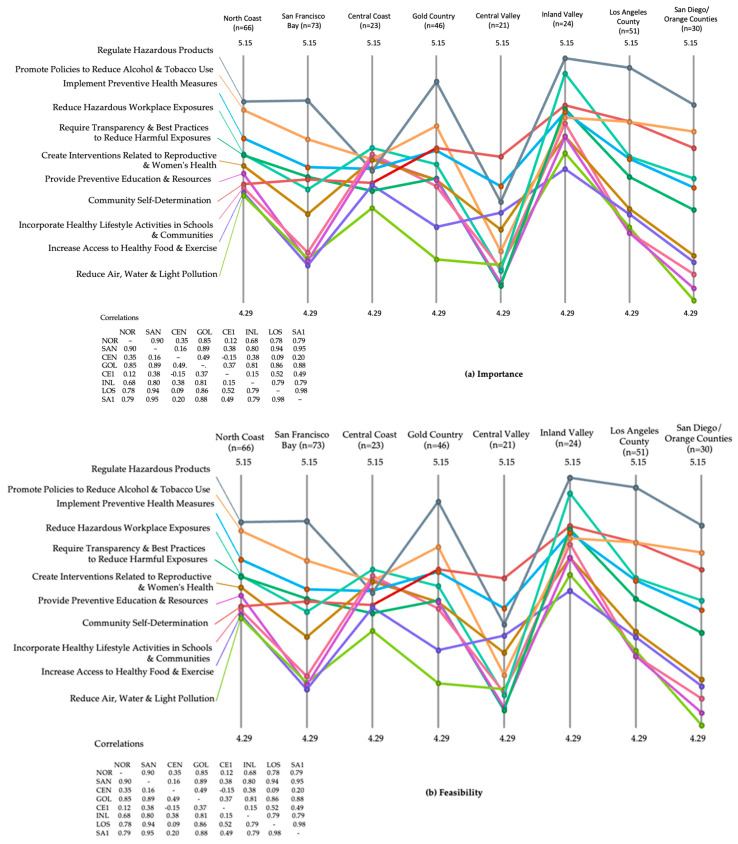
Ladder diagram by region of CA for (**a**) Importance and (**b**) Feasibility.

**Figure 7 ijerph-21-00318-f007:**
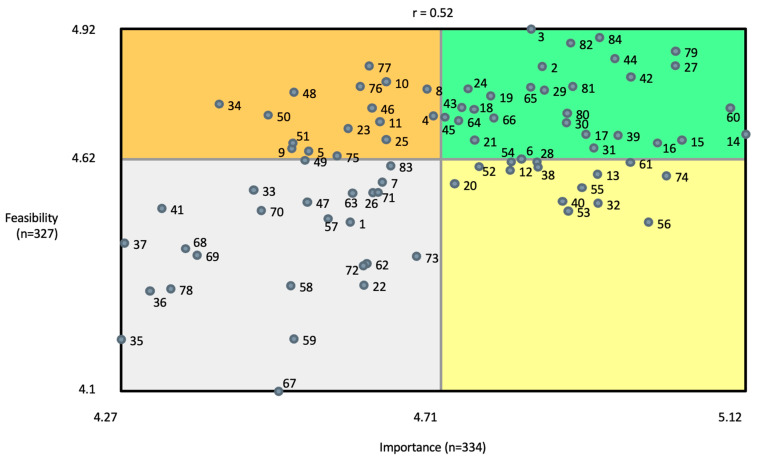
Go-zone map for all 84 potential interventions across all 11 clusters and including all participants.

**Figure 8 ijerph-21-00318-f008:**
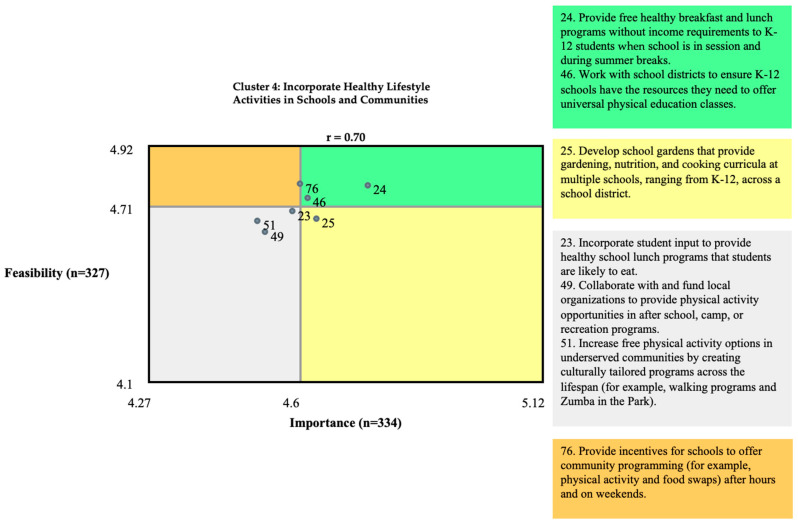
Go-zone for Cluster 4, including individual interventions.

**Table 1 ijerph-21-00318-t001:** Sample demographic characteristics of the 351 participants with complete data.

Sample Characteristic		Number (%)
Stakeholder category	Community member	194 (55.3)
	Healthcare practitioner	85 (24.2)
	Academic/Researcher	37 (10.5)
	Policy expert/Advocate	34 (9.7)
	No response	1 (0.3)
Region of California	Northern CA	69 (19.7)
	San Francisco Bay Area	75 (21.4)
	Central Coast	28 (8.0)
	Gold Country	49 (14.0)
	Central Valley	24 (6.8)
	Inland Valley (Imperial Valley)	25 (7.1)
	Los Angeles County	51 (14.5)
	San Diego and Orange Counties	30 (8.6)
Race/ethnicity	White	191 (54.4)
	Black/African American	66 (18.8)
	Latina/Hispanic/Latinx	48 (13.7)
	American Indigenous/Indian/Native Alaskan	13 (3.7)
	Asian	13 (3.7)
	Native Hawai’ian/Pacific Islander	8 (2.3)
	Bi- or multi-racial	7 (2.0)
	No response	5 (1.4)
Age (years)	≤25	0 (0)
	26–40	224 (63.8)
	41–59	94 (26.8)
	60+	22 (6.3)
	No response	11 (3.1)
Gender identity	Female/Woman	275 (78.4)
	Male/Man	66 (18.8)
	Gender non-conforming (non-binary, gender queer, gender fluid)	2 (0.6)
	Transgender	0 (0.0)
	No response	8 (2.3)
Living with a diagnosis of breast cancer?	Yes	94 (26.8)
	No	256 (72.9)
	No response	1 (0.3)

## Data Availability

A synopsis of data, broken down by cluster and individual interventions, is presented in Appendix A.

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
