# Peer review of "Development of Systemic Interventions to Decrease Breast Cancer Risk: A Group Concept Mapping Study"

_ijerph, 2024, doi:10.3390/ijerph21030318_

Round 1

Reviewer 1 Report

Comments and Suggestions for Authors

The manuscript is lengthy.

Some parts may be shortened to reduce the length.

Remove Table 1 from the manuscript, it should be included only the risk factors included in the paper.

The aim and objectives of the study should be described more clearly.

The methods are confusing because they do not allow a clear understanding of what was done.

Please include a flow diagram of the selection process with the final number of individuals assessed.

What is the importance of including section "2.1 Advisory group".

Maybe some parts of the methods section could be moved to an appendix or to the supplementary materials.  

The study design is not clearly stated because it is mixed within all the sections.

The final sample size can not be easily found within the large amount of information presented in the Methods section.

Several parts of the manuscript must by synthesized to reduce its length.

Including the statistical analyses mixed within the description of the methodological framework causes confusion and no reference are presented to validate the choice of each model or statistical tool. Likewise, nothing is said regarding the software and the characteristics of the model. For instance, 

Authors should address some major issues within the manuscript before being accepted for publication in the journal. 

Author Response

REVIEWER 1

Thank you for your careful reading of our manuscript and your suggestions for how we could improve it.  We believe that responding to your concerns has made it a stronger and more readable text.

The manuscript is lengthy. 

We have carefully revised the full manuscript to remove redundancies and to shorten sections wherever this did not interfere with the core information of the paper.  This process led to deletions and edits throughout the manuscript.

On the other hand, each reviewer requested additional information be added resulting, in some sections, in slight expansion of the text.

Some parts may be shortened to reduce the length.

See response above.

Remove Table 1 from the manuscript, it should be included only the risk factors included in the paper.

We have removed Table 1 from the manuscript and added only the 15 risk factors included in the current study into the text in the lines at the end of the third paragraph of section 1.2.

The aim and objectives of the study should be described more clearly.

In our reorganization of the manuscript, we now have a separate small section (1.3) on ‘aims ad objectives of the current project’.    

The methods are confusing because they do not allow a clear understanding of what was done.

We have reorganized and abbreviated the Methods and Materials section, removing all ‘results’ materials into the Results section.  We have also added a Methods Flow Chart (new Figure 1) which briefly summarizes the process of intervention selection, as well as the overall study structure including the number of participants at each step.

Please include a flow diagram of the selection process with the final number of individuals assessed.

See response above.

What is the importance of including section "2.1 Advisory group".

We have included section 2.1 which describes the Advisory Group because they were an essential part of the project from its inception.  We briefly describe their selection and roles in this section and then refer to specific tasks they undertook in other places in the Methods section.

Maybe some parts of the methods section could be moved to an appendix or to the supplementary materials.  

Thank you for this suggestion.  We believe that the reviewer was concerned both about the length of the Methods section and the confusion which appears to have arisen for two of the reviewers given the inclusion of the statistical results within the Methods section.

As described above, we have addressed this issue by creating a Methods Flow Chart and moving all results, including the statistics on validity and reliability of the process, to the Results section.  There we have shortened the statements, as possible, to keep in the key information while having the prose of the manuscript flow more smoothly.

The study design is not clearly stated because it is mixed within all the sections.

See comments above on Methods Flow Chart and moving the text to clarify the distinction between the Methods and the Results.

The final sample size cannot be easily found within the large amount of information presented in the

Methods section.

We have clarified the sample size withing the text (second paragraph of section ‘2.3 Participants’) as well as in the Methods Flow Chart.

Several parts of the manuscript must by synthesized to reduce its length.

Including the statistical analyses mixed within the description of the methodological framework causes confusion and no reference are presented to validate the choice of each model or statistical tool.

See responses above.

Likewise, nothing is said regarding the software and the characteristics of the model. For instance,

We have added information about the groupwisdomTM software and the framework of the Group Concept Mapping model, complete with references, to new section 2.4.

Reviewer 2 Report

Comments and Suggestions for Authors

1. Page 4 of 20

Line number : 113-120.

Authors may need to consider providing a supplementary flowchart that presents the number of intervention possibilities decreasing from 141 to 84, along with listing the exclusion criteria as described in the paragraph.

2. Page 5 of 20 

Line number : 190-200

It might be beneficial for authors to provide the advantages of cluster analysis and explain why conducting such an analysis is necessary for the study.

3. Page 8 of 20; 9 of 20; 10 of 20

Figure 1-2

I understand that Figures 1-2 display cluster results, but authors may need to reconstruct or modify the figures slightly to enhance clarity. Currently, interpreting them is somewhat challenging due to the overlap of the figure and legend, as indicated in the figures.

4. Page 17 of 20

The strength of the current project is well written; however, it would enhance the completeness of the discussion if authors also addressed the potential limitations of their project.

Comments on the Quality of English Language

English is acceptable. Minor editing of English language required

Author Response

REVIEWER 2

Thank you for your supportive suggestions.  We believe that responding to your concerns has made it a stronger and more readable manuscript.

  1. Page 4 of 20

Line number : 113-120.

Authors may need to consider providing a supplementary flowchart that presents the number of intervention possibilities decreasing from 141 to 84, along with listing the exclusion criteria as described in the paragraph.

We have added a Methods Flow Chart (new Figure 1) which briefly summarizes the process of intervention selection, including the exclusion criteria.

  1. Page 5 of 20 

Line number : 190-200

It might be beneficial for authors to provide the advantages of cluster analysis and explain why conducting such an analysis is necessary for the study.

We have rewritten the final paragraph of the Introduction to introduce Group Concept Mapping (GC) very briefly.  In that sentence we begin to clarify that the very nature of the GCM framework includes,synthesizing what is learned into a set of common themes or activities that can then be put into action [14,15].”  In the Methods and Materials section, we have added a more explicit description of GCM: “GCM is a mixed methods framework that first gathers qualitative data in the form of a large number of responses or statements to an open-ended question [14]. Multivariate analyses, consisting of multidimensional scaling [16] and hierarchical cluster analysis [14,16,17], are then performed within the groupwisdomTM software which produces visuals including cluster maps, pattern matches (ladder diagrams), and go-zones (see below) that aid in analysis, interpretation, and utilization [14,15].”

Hopefully this makes clear that the hierarchical cluster analysis is an integral part of the GCM methodology.  We have provided numerous citations to articles describing the development of GCM and its use as an analytic framework for projects with similar goals to ours.

  1. Page 8 of 20; 9 of 20; 10 of 20

Figure 1-2

I understand that Figures 2-3 (current numbering, with addition of the requested Methods Flow Chart) display cluster results, but authors may need to reconstruct or modify the figures slightly to enhance clarity. Currently, interpreting them is somewhat challenging due to the overlap of the figure and legend, as indicated in the figures.

We have redesigned both Figures 2 and 3 (current numbering, with addition of the requested Methods Flow Chart) so that the text within the figures does not overlap with the geometric figures.  Additionally, we have now presented the data in Figure 3 (new numbering) in the same color-coded system as was used in Figure 2.  This should make initial reading, comparisons and interpretations easier for the reader.

  1. Page 17 of 20

The strength of the current project is well written; however, it would enhance the completeness of the discussion if authors also addressed the potential limitations of their project.

Thank you for this suggestion.  We have added a separate section (4,2) in the Discussion for ‘Strengths and limitations of the project’.  We have incorporated into that section edited versions of materials that already existed, addressing the strengths of the work.  We have also added comments on limitations of the project.

Reviewer 3 Report

Comments and Suggestions for Authors

Thank you for giving me the opportunity to review this paper. The manuscript well-written based on the studied topic. I have few minor comments and suggestions for authors:

1. Subsection 2.5.1 - validity and reliability; please add the standard cut-offs defined as strong, moderate or weak with references.

2. It would be good to have a subsection under discussion - implications for research, policy and practice from the current research impact

3. Have a separate sub-section on strengths and limitations of the study.

4. Figure 2 needs some editing. In its current form, the words related to the figure is not visible.

Comments on the Quality of English Language

English language is fine.

Author Response

REVIEWER 3

Thank you for giving me the opportunity to review this paper. The manuscript well-written based on the studied topic.

Thank you for your careful reading of the paper and for your supportive comments and suggestions.

 I have few minor comments and suggestions for authors:

1.Subsection 2.5.1 - validity and reliability; please add the standard cut-offs defined as strong,

moderate or weak with references.

We very much appreciate the reviewer’s close reading of the ‘validity and reliability’ section, and the request for cut-offs, bracketed limits that are often presented with statistics used to measure the strength of data within a certain analytic framework.  However, for GCM, there are no accepted cut-offs for weak, moderate or strong coefficients, but there are accepted mean values established through two meta-analyses now cited in an introductory sentence in the reliability/validity section. We believe this is responsive to this comment. 

  1. It would be good to have a subsection under discussion - implications for research, policy and practice from the current research impact. Thank you for this suggestion. Although another reviewer has asked us to shorten the manuscript, if possible, we believe that the Discussion has been strengthened with the reorganization and addition of new materials in the new section titled, ‘4.3 Moving forward: Implications for research, policy and practice.’

  1. Have a separate sub-section on strengths and limitations of the study.

Thank you for this suggestion.  We have added a separate section (4,2) in the Discussion for ‘Strengths and limitations of the project’.  We have incorporated in that section (edited versions of) materials that already existed, addressing the strengths of the work.  We have also added comments on limitations of the project.

  1. Figure 2 needs some editing. In its current form, the words related to the figure is not visible.

We have redesigned both Figures 2 and 3 (current numbering, with addition of the requested Methods Flow Chart) so that the text within the figures does not overlap with the geometric figures.  Additionally, we have now presented the data in Figure 3 (new numbering) in the same color-coded system as was used in Figure 2.  This should make initial reading, comparisons, and interpretation easier for the reader.

Round 2

Reviewer 1 Report

Comments and Suggestions for Authors

The authors have addressed all the comments and issues.